# Absolute Camera Pose Regression Using an RGB-D Dual-Stream Network and Handcrafted Base Poses

**DOI:** 10.3390/s22186971

**Published:** 2022-09-15

**Authors:** Peng-Yuan Kao, Rong-Rong Zhang, Timothy Chen, Yi-Ping Hung

**Affiliations:** 1Graduate Institute of Networking and Multimedia, National Taiwan University, Taipei 10617, Taiwan; 2Department of Computer Science & Information Engineering, National Taiwan University, Taipei 10617, Taiwan

**Keywords:** absolute camera pose regression, dual-stream network, handcrafted base poses

## Abstract

Absolute pose regression (APR) for camera localization is a single-shot approach that encodes the information of a 3D scene in an end-to-end neural network. The camera pose result of APR methods can be observed as the linear combination of the base poses. Previous APR methods’ base poses are learned from training data. However, the training data can limit the performance of the methods, which cannot be generalized to cover the entire scene. To solve this issue, we use handcrafted base poses instead of learning-based base poses, which prevents overfitting the camera poses of the training data. Moreover, we use a dual-stream network architecture to process color and depth images separately to get more accurate localization. On the *7 Scenes* dataset, the proposed method is among the best in median rotation error, and in median translation error, it outperforms previous APR methods. On a more difficult dataset—*Oxford RobotCar* dataset, the proposed method achieves notable improvements in median translation and rotation errors compared to the state-of-the-art APR methods.

## 1. Introduction

Single-shot camera localization (also known as the model-based localization, image-based localization, visual localization, or camera relocalization) estimates the camera pose from a single image related to a previously visited scene. It has two phases. In phase 1, a model (e.g., 3D point cloud, neural network) is first built based on a set of images. In phase 2, the camera pose related to the model is estimated based on a single image. Single-shot camera localization can be divided into six categories: the *Structure-Based Method* [1], *Structure-Based Using Image Retrieval* [2], *Pose Interpolation* [3,4], *Relative Pose Estimation* [5,6], *Scene Point Regression* [7,8,9], and *Absolute Pose Regression* [10,11,12,13,14,15].

Assume that in phase 1, the scene model has been built. In Phase 2, for the *Structure-Based Method* [1], the first step is to extract local features of the query image. The second step is 2D-3D feature matching of the query image and the model. After obtaining the 2D-3D correspondences, the third step is to estimate the camera pose by the PnP algorithm. The *Structure-Based Using Image Retrieval* method [2] is similar to the *Structure-Based Method*. The difference between them is that the *Structure-Based Using Image Retrieval* method uses the image retrieval method to reduce the search region of 2D-3D feature matching, which can make the method estimate fast in a large environment. The *Pose Interpolation* method [3,4] is to find the top *k* similar images of the query image in the reference database and interpolate the camera poses of these retrieved images to obtain the pose estimate of the query image. After image retrieval, the *Relative Pose Estimation* method [5,6] employs a relative pose regression (RPR) to achieve more accurate localization. The query image and nearest neighbor reference image are simultaneously input into the RPR network. The task of this type of RPR network is to learn both the relative pose between the retrieved image pairs and the query image and the global features of a single image, regardless of the scene. The *Scene Point Regression* method [7,8,9] is similar to the *Structure-Based Method*. The difference between them is that the *Scene Point Regression* no longer relies on feature extraction to conduct 2D-3D feature matching. It uses the machine-learning method or deep-learning method to regress the correspondences of 2D image points and 3D model points. The *Absolute Pose Regression* method [10,11,12,13,14,15] uses an end-to-end deep-learning model to regress the camera pose of the query image.

Among these six categories, the *Absolute Pose Regression* (APR) method uses an end-to-end deep-learning model to regress the camera pose of the query image quickly and needs less storage to store the neural network model. However, the localization accuracy of the APR method is not as good as the other methods. Moreover, the camera pose result of APR methods can be observed as the linear combination of the base poses [16]. The previous APR methods’ base poses are learned from training data. APR methods are a data-driven camera localization approaches by which the distribution of the ground truth camera trajectories used for training can limit base poses learned from the training data. As long as three orthonormal vectors exist, their linear combination can cover the entire 3D space, as the number of base poses is often much more significant than three, and learning-based base poses are redundant. However, since the training trajectory does not necessarily include diverse directions, the redundant base poses may not cover the space outside the training trajectory. In other words, the APR model that adopts base poses based on learning might have no generalizability. To reduce the impact of this issue, we turn to handcrafted base poses instead of using learning-based base poses.

In this paper, we propose an absolute pose regression (APR) method for camera localization using an RGB-D dual-stream network and handcrafted base poses. This method fuses color and depth information for more accurate localization. We use a dual-stream network architecture to process color images and depth images separately and combine this with handcrafted base poses to reduce the impact of the network’s limitation to movement trajectories in the training data.

We conducted detailed ablation studies, which demonstrate the efficacy of the proposed method. Furthermore, comprehensive evaluation results demonstrate that on the *7 Scenes* dataset, the proposed method is among the best in the median rotation error, and in the median translation error, it outperforms previous APR methods. On more difficult dataset—*Oxford RobotCar* dataset, the proposed method achieves notable improvements in median translation and rotation errors, as compared to the state-of-the-art APR methods.

To summarize, this work has three contributions:We use handcrafted base poses instead of learning-based base poses (e.g., [12,13,14,15]) to estimate the camera poses, which can prevent the camera poses from overfitting the training data’s camera trajectories.We use a dual stream network to fuse color and depth information to obtain more accurate localization results.On the *7 Scenes* dataset, the proposed method is among the best in the median rotation error, and in the median translation error, it outperforms previous APR methods. On more difficult dataset—*Oxford RobotCar* dataset, the proposed method achieves notable improvements in median translation and rotation errors, as compared to the state-of-the-art APR methods.

The remainder of this paper is organized as follows. In Section 2 we review related works, and in Section 3 we describe the proposed APR method. Section 4 presents the experimental results to attest to the effectiveness of the proposed method, and Section 5 concludes this study.

## 2. Related Work

In this section, we will focus on the related works of the learning-based single-shot camera localization approaches.

### 2.1. Structure-Based Method

For *Structure-Based Method* [1], the first step is to extract local features of the query image. The second step is 2D-3D feature matching of the query image and the model. After obtaining the 2D-3D correspondences, the third step is estimating the camera pose using the PnP algorithm.

### 2.2. Structured-Based Using Image Retrieval

Structure-based localization using a 3D map can lead to higher accuracy. However, as the scale of the scene increases, the number of key points increases, leading to increased memory requirements and longer run times for feature matching. This also produces more wrong 2D-3D matches, leading to large localization errors. Therefore, HF-Net [2] adopts coarse-to-fine localization to handle large-scale environments. First, image retrieval is leveraged to find the nearest neighbors of the query image, reducing the search space for subsequent matching. Then, a structure-based pipeline is applied to retrieved images to estimate an accurate and robust camera pose.

### 2.3. Pose Interpolation

Camera localization based on image retrieval [4] involves finding the top *k* similar images of the query image in the reference database and interpolating the camera poses of these retrieved images to obtain the pose estimate of the query image.

### 2.4. Relative Pose Regression

For a more accurate localization, relative pose regression (RPR) [5,6] can be used after image retrieval. The query image and its nearest neighbor reference image are simultaneously input into the RPR network. The task of this type of RPR network is to learn the relative pose between the retrieved image pairs and the query image, and the global features of a single image, regardless of the scene. Thus, these methods can be trained on a general scene or multiple scenes.

### 2.5. Scene Point Regression

Scene coordinate regression no longer relies on feature extraction but implements the first stage through machine learning or deep learning. SCRF [7] employs random forests to regress the global 3D points corresponding to the 2D points in the image. To train in an end-to-end manner, differentiable RANSAC (DSAC) [8] uses one network for scene coordinate regression and another for scoring regression. DSAC++ [9] solves the shortcomings of DSAC requiring a 3D model during training and counts soft inliers to select the best camera pose hypothesis instead of a scoring network.

### 2.6. Absolute Pose Regression

PoseNet [10] is the first method to directly regress 6-DoF camera poses from a single RGB image in an end-to-end manner. Based on the network architecture of PoseNet’s network architecture, the Bayesian PoseNet [11] adds dropout layers after FC (fully-connected) layers of position and rotation regressors. Without regard to fine-tuning the hyperparameters of absolute position loss and absolute rotation loss, PoseNet-17 [12] proposes absolute pose loss with learnable weights. Li’s method [13] is the first work to regress camera poses with a dual-stream network, taking color and depth images as the input. They propose an encoding method, minimized normal + depth (MND), to provide more information when expanding raw depth images into three-channel images. MapNet [14] achieves better performance by using the logarithm of a unit quaternion instead of the quaternion as the rotation formalism. Also proposed with MapNet, are MapNet+ and MapNet+PGO. Based on a CNN trained by MapNet, MapNet+ further updates model parameters in a self-supervised manner. In learning from unlabeled data, sensor measurements such as the GPS, IMU, or relative camera pose are calculated using a visual odometry algorithm as the pseudo ground truth for the relative pose. During the inference stage, MapNet+PGO feeds the camera poses of consecutive frames predicted by MapNet+ into the pose graph optimization (PGO) algorithm for optimization to obtain a smooth predicted pose trajectory. MapNet minimizes the absolute pose loss of a single image and the relative pose loss between image pairs in terms of the loss function. The model implicitly learns the visual odometry information. Tian’s method [15] also expects the model to understand the motion between consecutive frames. In addition to the typical absolute pose loss, pixel-level photometric loss and image-level structured similarity (SSIM) loss are introduced as geometric reconstruction constraints. Therefore, depth image and prediction of the relative pose are leveraged to obtain the reconstructed image by 3D warping during the training process.

According to [16], camera pose result of APR methods can be observed as the linear combination of the base poses. Previous APR methods’ base poses are learned from training data. However, the training data can limit the performance of the methods, which cannot be generalized to cover the entire scene. In this paper, we use handcrafted base poses instead of learning-based base poses, which prevents the overfitting of the camera poses of the training data.

## 3. Proposed Method

### 3.1. Notation

Let $p_{t}=\left(c_{t},r_{t}\right)$ be the ground-truth camera pose of the *t*-th image $I_{t}\;\left(I_{t}^{\mathrm{RGB}}/I_{t}^{D}/I_{t}^{\mathrm{RGBD}}\right)$, where $c_{t}\in R^{3}$ is the camera position and $r_{t}\in R^{d_{r}}$ is the camera rotation. The $d_{r}$ symbol represents the dimension of the rotation parametrization. We use ${\hat{p}}_{t}=\left({\hat{c}}_{t},{\hat{r}}_{t}\right)$ to denote a camera pose predicted by absolute camera pose regression.

### 3.2. Theory of Absolute Camera Pose Regression

#### 3.2.1. Theoretical Model

As shown in Figure 1, we divide the network architecture of APR methods into three components based on [16].

The first part is to extract features from the input image. Usually, this involves using the classic CNN network architecture, e.g., GoogLeNet, VGG, or ResNet. Weights of a model initialize the weights of this part pre-trained on a large-scale image recognition dataset (such as ImageNet), so that the network can learn more general and rich feature representations. The second part aims to reduce over-fitting by mapping the learned feature into an embedding space. The third part is to linearly transform the features or embeddings to the space of the 6-DoF camera pose, which is implemented by FC (fully connected) layers. Specifically, two FC layers are used as regressors: one for camera position and the other for camera rotation. The output of the position regressor represents the 3D coordinates of the query image in the scene; thus, the output dimension of its FC layer is 3. However, the output dimension of rotation regressor $d_{r}$ depends on the rotation parametrization used. Most APR methods use a 4-d unit quaternion or 3-d logarithm of a unit quaternion to represent rotation.

Let $\left(x_{1},x_{2},\cdots ,x_{n}\right)$ and $\left(w_{1},w_{2},\cdots ,w_{n}\right)$ be the projection matrix of the translation part and orientation part, respectively, where $x_{j}\in R^{3}$ and $w_{j}\in R^{d_{r}}$ denote the *j*-th column. Their bias terms are represented by $x_{0}$ and $w_{0}$. Features extracted from image $I_{t}$ are denoted as $\alpha^{t}={\left(\alpha_{1}^{t},\alpha_{2}^{t},\cdots ,\alpha_{m}^{t}\right)}^{T}\in R^{m}$, and separate embeddings of feature $\alpha^{t}$ are denoted by $\lambda^{t}={\left(\lambda_{1}^{t},\lambda_{2}^{t},\cdots ,\lambda_{n}^{t}\right)}^{T}\in R^{n}$ and $\mu^{t}={\left(\mu_{1}^{t},\mu_{2}^{t},\cdots ,\mu_{n}^{t}\right)}^{T}\in R^{n}$. The camera pose predicted by APR can be expressed as
(1)$$\left(\begin{array}{c} {\hat{c}}_{t}\\ {\hat{r}}_{t} \end{array}\right)=\left(\begin{array}{c} x_{0}+\sum_{j=1}^{n}\lambda_{j}^{t}x_{j}\\ w_{0}+\sum_{j=1}^{n}\mu_{j}^{t}w_{j} \end{array}\right),$$

#### 3.2.2. Rotation Formalisms

APR approaches generally utilize the unit quaternion or rotation vector (the so-called axis-angle) to represent rotation. The unit quaternion has the advantage of being interpolable, so it is often the preferred approach to representing rotation in the field of computer vision. The arbitrary 4-d output of the regressor can be normalized and then used to represent legitimate 3-DoF rotation, but when used in training, this step has been demonstrated to degrade performance [10]. Therefore, APR methods that use the quaternion as the rotation formalism generally ignore the sphere constraint of the unit quaternion during training. Unlike the over-parameterized unit quaternion, the rotation vector requires a 3-d vector to describe rotation. Its direction and magnitude, respectively, represent the axis and angle of rotation. The conversion formula (logarithm) to express the unit quaternion q with the axis of rotation n and the angle of rotation θ is
(2)$$q\;=\left(u,v\right)=\left(\cos\frac{\theta }{2},n\sin\frac{\theta }{2}\right)=e^{n\cdot \frac{\theta }{2}}.$$

As can be observed from the above formula, this rotation representation is similar to the rotation vector. The only difference is that the length of the logarithm of a unit quaternion is equal to half of the rotation angle. The proposed method also uses the logarithm of a unit quaternion to represent the camera rotation.

#### 3.2.3. Base Poses

According to [16], we can regard ${\hat{c}}_{t}$ (or ${\hat{r}}_{t}$) as a linear combination of $x_{j}$ (or $w_{j}$) according to Equation (Equation 1). For a trained network, the parameters x (the ‘base translations’) and w (the ‘base orientations’) of the last two FC layers are unchanged and can be regarded as a set of base poses $B=\{\left(x_{j},w_{j}\right)\}$. The value of an embedding (or feature) related to the appearance of the input image can be viewed as the coefficients of base poses to measure their importance. Therefore, we consider that APR methods mainly learn the distribution of base poses and the correlation between the appearance of the image and base poses, so that the matrix product of embedding (or feature) and base poses can be used to estimate the camera pose for a single image.

APR is a data-driven camera localization approach, by which the distribution of the ground truth camera trajectories used for training can limit base poses learned from the training data. As long as three orthonormal vectors exist, their linear combination can cover the entire 3D space. As the input dimension of the last FC layer, the number of base poses is often much more significant than three, and the learning-based base poses are redundant. However, since the training trajectory does not necessarily include diverse directions, redundant base poses may not cover the space outside the training trajectory. In other words, the APR model that adopts base poses based on learning might have no generalizability. To reduce the impact of CNNs, which cannot generalize with limited training data, we turn to handcrafted base poses instead of using FC layers to learn base poses.

### 3.3. Network Architecture

The proposed dual-stream network with handcrafted base poses is designed for RGB-D single-shot camera localization. The overall network architecture is shown in Figure 2. The color stream (top) and the depth stream (bottom) have the same architecture but different weights. In each stream, we use a partially modified ResNet-18 as the network’s backbone. We discard its original FC and softmax layers designed for classification and instead use an FC layer with an output dimension of 2048. The global features of the image output by the modified ResNet-18 are subsequently mapped into the embedding space by two FC layers. One of the embeddings is viewed as the scale factors of the handcrafted base translations and the other as the scale factors of the handcrafted base orientations. A color image and its corresponding depth image are fed to the dual stream, after which every single stream outputs two embeddings. We fuse color and depth information by adopting the average of their embeddings as coefficient vectors of the handcrafted base poses. Finally, the predicted camera pose is obtained by matrix multiplication of the handcrafted base poses and the average coefficients.

### 3.4. Loss Function

As shown in Figure 3, the network model is expected to encode geometric information about the scene through the absolute pose loss of a single image and the relative pose loss between the temporally adjacent images.

The overall loss function is
(3)$$L=L_{abs}+L_{rel}=\sum_{t=1}^{\left|D\right|}h\left({\hat{p}}_{t},p_{t}\right)+\gamma \sum_{t=1}^{\left|D\right|}h\left({\hat{v}}_{t}^{t+\Delta t},v_{t}^{t+\Delta t}\right),$$
where $v_{t}^{t+\Delta t}=\left(c_{t}-c_{t+\Delta t},r_{t}-r_{t+\Delta t}\right)$ and $h\left(\hat{p},p\right)={∥\hat{c}-c∥}_{1}+\beta \cdot {∥\hat{r}-r∥}_{1}$, β, and γ are hyperparameters.

Relative pose loss is helpful for a network to learn the features suitable for localization. Transfer learning is employed by using a model pre-trained for image classification as a feature extractor. Image classification identifies the category to which the subject of an image belongs; thus, images that are similar in appearance also have similar features. When training to regress camera poses, the same subjects often appear in many input images, and the network must learn the transformation of poses from slight differences in similar appearances. We use the linear combination of base poses to predict the relative pose:(4)$${\hat{v}}_{t}^{t+\Delta t}=\left(\begin{array}{c} {\hat{c}}_{t}-{\hat{c}}_{t+\Delta t}\\ {\hat{r}}_{t}-{\hat{r}}_{t+\Delta t} \end{array}\right)=\left(\begin{array}{c} \sum_{j=1}^{n}\left(\lambda_{j}^{t}-\lambda_{j}^{t+\Delta t}\right)x_{j}\\ \sum_{j=1}^{n}\left(\mu_{j}^{t}-\mu_{j}^{t+\Delta t}\right)w_{j} \end{array}\right).$$

Note from Equation (Equation 4), that if the values of the base pose remains unchanged during training; then, the network focuses on learning the feature embedding difference of image pairs by minimizing the relative pose loss, which also takes advantage of the handcrafted base poses.

### 3.5. Training Mechanism

We adopt a two-stage training mechanism for the proposed RGB-D dual-stream network. In the first stage, as the color stream and depth stream are independent networks, there is no longer an operation to calculate the average value of their feature embeddings shown in Figure 2. Separate feature embedding is multiplied directly by handcrafted base poses. As a result, two predictions are obtained, one of which is related only to color information, and the other is related only to depth information. The first stage’s goal is to allow two streams to converge separately in preparation for the next stage. The second stage is the end-to-end training of the overall RGB-D dual-stream network. The dual-stream network better integrates color and depth features to achieve more accurate camera localization by fine-tuning the weights.

### 3.6. Implementation Details

Before feeding RGB-D images to the proposed dual-stream network, we preprocess the images. Compared with color image preprocessing, there are two more operations for depth images: depth completion and stacking (see Section 3.6.1 for details). Both color and full depth images are resized to 341 × 256 pixels and then subtracted by their mean image and divided by standard deviation. We use Adam optimization with a learning rate of 1 × $10^{-4}$, a weight decay of 5 × $10^{-4}$, and the default values for the other parameters. With a mini-batch size of 10, we train two separate streams for 300 epochs in the first stage and train the overall dual-stream network for 100 epochs in the second stage. The β and Δt hyperparameters are set to 3 and 10 for all scenes, and the γ is set to 10 for indoor and 1 for outdoor datasets. The algorithm was implemented based on PyTorch, and all experiments were performed on an NVIDIA GTX 1080Ti graphics card.

#### 3.6.1. Depth Completion

As demonstrated in Figure 4a, invalid depth values sometimes occur when using RGB-D sensors such as Kinect to capture depth images. To maintain the performance of the depth stream, we impute and fill in these depth values through a modified colorization method [17], which was used in the *NYU Depth Dataset V2* [18] for the same purpose.

This depth completion optimization assumes that those adjacent pixels with similar intensity have similar depth. Missing values are imputed by minimizing the difference between the depth value at pixel r and the weighted sum of depth values at its neighbors N(r):(5)$$\begin{array}{c} \min_{F}\sum_{r}{\left(F\left(r\right)-\sum_{s\in N(r)}w_{\mathrm{rs}}F\left(s\right)\right)}^{2},\\ \mathrm{subject}\;\mathrm{to}\;F\left(v_{i}\right)=D\left(v_{i}\right),\forall v_{i}\in \left\{v|D\left(v\right)\;\mathrm{is}\;\mathrm{valid}\right\}, \end{array}$$
where *D* and *F* separately denote the raw and full-depth image, and $w_{\mathrm{rs}}$ is the similarity measurement between intensity at pixel r and intensity at pixel s: $w_{\mathrm{rs}}=\exp\left(-\frac{{\left(G(r)-G(s)\right)}^{2}}{2\sigma_{r}^{2}}\right)$.

After depth completion, it is necessary to expand the single-channel depth images into three-channel images via stacking to obtain the final input of the feature extractor.

#### 3.6.2. Selection of Handcrafted Base Poses

Based on the discussions of base poses in Section 3.2.3, we select six orthonormal vectors to represent base poses. To ensure the non-linearity, we choose not to adopt a minimum of three base poses of feature embedding. If only three orthonormal vectors in the positive direction are used as base poses, to guarantee that the calculated pose estimate covers the entire 3D space, the value range of the feature embedding—or the coefficients of base poses—should be the entire set of real numbers. This requirement can only be met by mapping the features to a linear embedding without an activation function. The linear transformation can be combined with the matrix multiplication of handcrafted base poses, resulting in the network learning the distribution of base poses. This results in six orthonormal vectors (three in positive and three in negative directions) used as handcrafted base poses. We adopt the leaky rectified linear unit (leaky ReLU) as the activation function for nonlinear embeddings to streamline network convergence.

We also designed two sampling methods for redundant handcrafted base poses. Figure 5b shows uniform sampling along the three coordinate axes *x*, *y*, and *z*, which is in line with the human understanding of translation. Figure 5c shows uniform sampling along the surface and the radius of a sphere, which is closer to the definition of the logarithm of a unit quaternion. The coordinates of each point in Figure 5 are viewed as the three values of a base translation or a base orientation. The direction and magnitude of a base translation respectively represent its moving direction and distance. In the same way, the direction and magnitude of a base orientation represent its rotation axis and half of its rotation angle. We compare the different base poses in detail in the next section.

## 4. Experiments

### 4.1. Datasets

To compare the proposed method with previous APR methods, we evaluated it on the *7 Scenes* [7] indoor dataset, a well-known benchmark for single-shot camera localization. The *7 Scenes* dataset contains RGB-D camera frames of seven small-scale indoor scenes captured by a Kinect sensor. For each scene, several sequences are recorded and split into the training and test sets. The ground-truth camera poses and 3D models are obtained using an implementation of KinectFusion [19,20]. Note that the 3D model is not used in our proposed method. Details about *7 Scenes* are provided in Table 1.

*12 Scenes* [21] is a slightly larger RGB-D indoor dataset, similar to *7 Scenes*. Several sequences of twelve rooms were captured by a Structure Core sensor coupled with an iPad camera. The researchers used the VoxelHashing framework [22] combined with global bundle adjustment to obtain ground-truth camera poses. For each scene, we used the first sequence as the training set and all other sequences as the test set. Since previous APR methods were not evaluated on this dataset, we employed the *12 Scenes* dataset only in the ablation studies. Details about *12 Scenes* are provided in Table 2.

*Oxford RobotCar* [23] is a large-scale outdoor dataset for long-term autonomous driving. The researchers repeatedly traversed the same route of approximately 10 km over a year. Various sensors were mounted to the vehicle: a trinocular stereo camera, three monocular cameras, two 2D LIDARs, a 3D LIDAR, and an inertial and GPS navigation system. This yielded the *Oxford RobotCar* dataset, which captures a wide range of variations in appearance and structure, such as illuminance changes, seasonal changes, dynamic objects, and roadwork. The proposed method was evaluated on the *Loop* route with a total length of 1120 m, and the fused GPS+Inertial data were used as ground-truth camera poses.

### 4.2. Comparison with the State-of-the-Art Methods

Table 3 shows the quantitative comparison of median localization error on *7 Scenes* with the state-of-the-art APR methods, including PoseNet-17 [12], Li’s method [13], MapNet [14], and Tian’s method [15]. The proposed method is very close to Tian’s method in terms of the median translation error and outperforms all methods in terms of the median rotation error.

For the experiment on the *Oxford RobotCar* dataset, the color stream with handcrafted base poses was used to evaluate the performance of the proposed method, as depth information captured by a 2D LIDAR scanner is sparse. We used the 2014-05-14-13-59-05 sequence for training and the 2014-05-14-13-53-47 sequence for testing. To account for illuminance variation between training and test sequences, we augmented the data by randomly changing images’ brightness, saturation, contrast, and hue during training. The result of PoseNet-15 in Table 4 was reproduced by Tian [15] using the same network and settings as that in [10]. Compared to Tian’s method [15], the proposed method reduces the median translation error by about 36% and the median rotation error by about 55%.

### 4.3. Ablation Studies

We conducted ablation experiments to explore the effects of depth completion, handcrafted base poses, and the dual-stream network.

We tested four types of base poses: one for learning-based base poses and three for handcrafted base poses: (a) *FC*, where the base poses are learned from the last two FC layers; (b) *Min*, where both base translations and orientations are orthonormal vectors; (c) *Cube*, where both base translations and orientations are uniformly sampled on the cube; (d) *Mixed*, where base translations are uniformly sampled on the cube and base orientations are uniformly sampled on the sphere. The number of base poses for *FC*, *Cube*, and *Mixed* was set to 2048, and that for *Min* was set to 6.

The network architectures evaluated in the ablation studies can be divided into three types: single-stream with learning-based base poses (Figure 6a), single-stream with handcrafted base poses (Figure 6b), and dual-stream with handcrafted base poses (Figure 2). Single-stream uses only color or depth images as input, whereas dual-stream uses RGB-D images. Note that if there are no extraordinary descriptions, the following depth stream, by default, takes the full depth images obtained by the depth completion algorithm as input.

#### 4.3.1. Effect of Depth Completion

Table 5 shows the difference in median localization the error between a single stream with raw depth images as the input and a single stream with full depth images obtained by the depth completion algorithm as the input.

The results demonstrate that the depth completion operation improves performance using learning-based or handcrafted base poses. When depth completion and handcrafted base poses are used simultaneously, the median localization error of the depth stream is significantly reduced.

#### 4.3.2. Comparison between Base Poses

To select the best set of handcrafted base poses to help the network learn features suitable for localization, we trained the color stream and depth stream with three different sets of handcrafted base poses. Single streams with learning-based base poses were also trained to evaluate the effect of handcrafted base poses. As shown in Table 6, *Min* outperforms the other two sets of redundant base poses. If base poses are redundant, one camera pose can be represented by various linear combinations of base poses. As *Min* is the most compact form of handcrafted base poses, the correlation between base poses is relatively weak. It may be easier for the network to learn the relationship between the appearance of images and these base poses.

We also observe from Table 6 that regardless of which set of base poses is adopted, the depth stream outperforms the color stream in the *Heads* scene. As reported in Table 1, the *Heads* scene has the smallest average valid depth of the frames. Depth images captured at close range are more likely to provide sufficient depth variation and structural information, which is exactly what the depth stream needs. Empirical experience suggests that if the scene is mostly flat and there are no complex 3D objects, the depth values captured by the depth sensor will be inaccurate, which may be why the results of the depth stream are inferior to those of the color stream in scenes such as *Chess*, *Pumpkin*, and *Stairs*.

To confirm this, we evaluated the performance of the color stream and depth stream on the *12 Scenes* dataset, whose average valid depth is close to that of the *Heads* scene. Table 7 shows that the median rotation error of the depth stream is close to that of the color stream, and the median translation error of the depth stream is smaller than that of the color stream. This shows that the average valid depth value and the presence of complex 3D objects in the scene limit the performance of the depth stream.

#### 4.3.3. Comparison between Network Architectures

*Dual stream* in Table 8 and Table 9 represents the proposed method using two-stage training. The performance of the dual-stream network on *7 Scenes* is slightly better than using only the color stream or depth stream. The results on *12 Scenes* demonstrate that the performance of the dual-stream network is inferior to the single depth stream in median translation error but is the best in median rotation error.

## 5. Conclusions & Future Work

We propose a novel APR method using an RGB-D dual-stream network and handcrafted base poses. On the *7 Scenes* dataset, the proposed method is among the best in median rotation error, and in the median translation error, it outperforms previous APR methods. On the more difficult datase—the *Oxford RobotCar* dataset, the proposed method achieves notable improvements in median translation and rotation errors, as compared to the state-of-the-art APR methods. The main contribution of this work is the use of handcrafted base poses instead of learning-based base poses (e.g., [10,11,12,13,14,15]) for estimating camera poses, which prevents overfitting to the camera poses of the training data.

To further improve the proposed method, a depth completion network (such as PENet [24], NLSPN [25], and FusionNet [26]) could be added to the depth stream to achieve end-to-end training. In addition, we could use the weighted sum of embeddings instead of the average value for the base pose coefficients to integrate better the feature embeddings of the color stream and depth stream. The embedding weights could be obtained by learning the confidence of each stream.

If 3D point clouds are available, it may be better to use these as input instead of depth images and adopt a network designed for unordered point sets (such as PointNet [27] and PointNet++ [28]) to extract the features. There are three advantages to using 3D point clouds. First, compared to depth images, point clouds provide richer structural information. Second, since sparse point cloud data can be directly fed into the network, there is no need for depth completion, which can improve localization efficiency. Third, 3D LiDAR is usually adopted for distance measurement in outdoor scenes, so using point clouds as input can remove the dual-stream network’s limitation to indoor scenes.

## Figures and Tables

**Figure 1 sensors-22-06971-f001:**
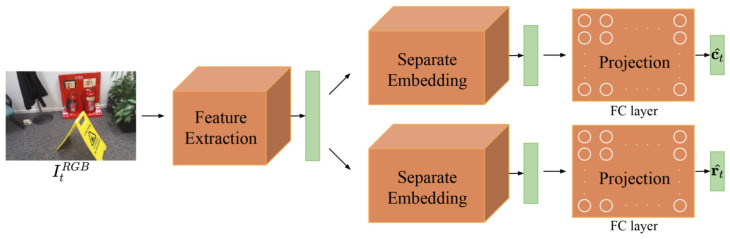
Network architecture of APR methods. Orange rectangles or blocks represent layers with learnable parameters, and green rectangles denote the output of each layer.

**Figure 2 sensors-22-06971-f002:**
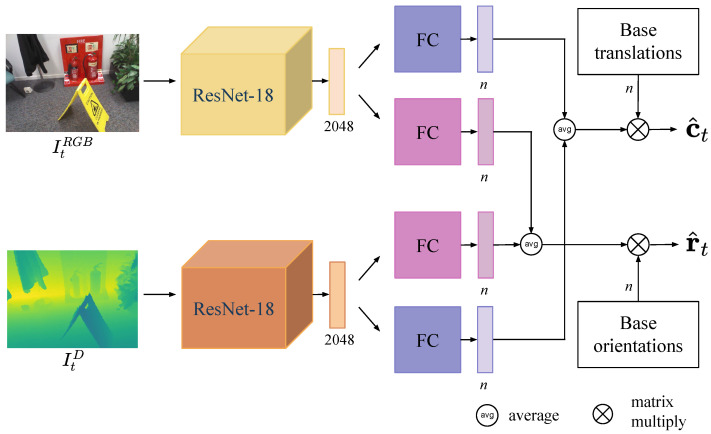
Network architecture of proposed method. Rectangles behind the ResNet-18 or FC layers represent their output vectors. Symbol *n* denotes the number of base poses as well as the output dimension of the FC layers.

**Figure 3 sensors-22-06971-f003:**
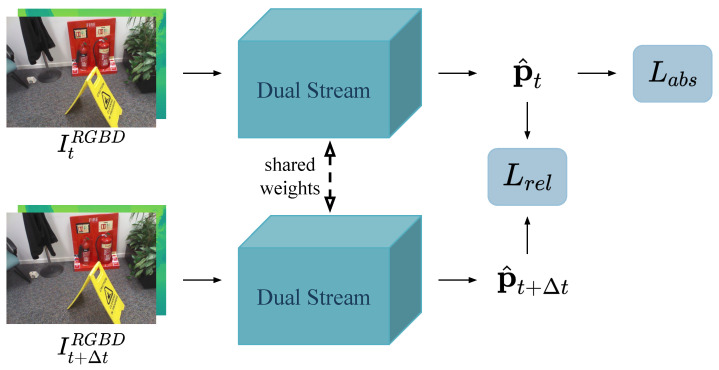
Training flow of proposed method.

**Figure 4 sensors-22-06971-f004:**
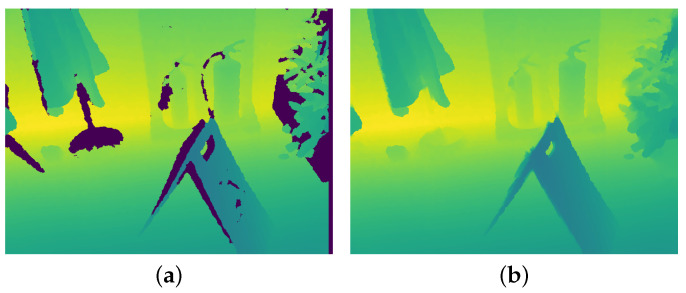
Comparison of a raw depth image (**a**) and its full depth image (**b**). The raw depth image is the first frame of the *Fire* scene from the *7 Scenes* dataset [7], and the full depth image is the result of depth completion.

**Figure 5 sensors-22-06971-f005:**
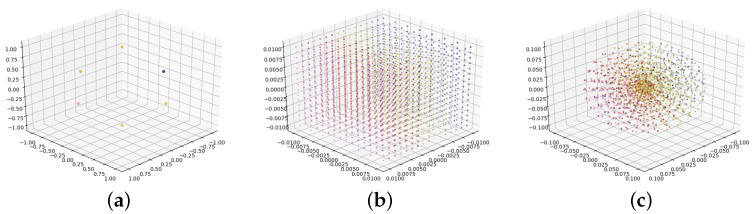
Three examples of handcrafted base poses. (**a**) Orthonormal vectors (n=6); (**b**) Uniform sampling in the cube (n=2048); (**c**) Uniform sampling in the sphere (n=2048). Symbol *n* denotes the number of handcrafted base poses. Different colors mean different handcrafted base poses.

**Figure 6 sensors-22-06971-f006:**
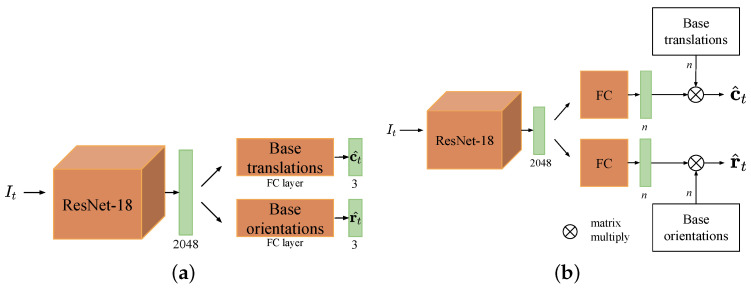
Two network architecture types for a single stream. (**a**) Single-stream with learning-based base poses; (**b**) single-stream with handcrafted base poses.

**Table 1 sensors-22-06971-t001:** Characteristics of *7 Scenes* dataset.

Scene	Frames		Volume	Ratio of Valid Depth		Mean of Valid Depth	
	Train	Test		Train	Test	Train	Test
Chess	4000	2000	6 m${}^{3}$	80.7%	86.0%	181 cm	170 cm
Fire	2000	2000	2.5 m${}^{3}$	89.5%	86.6%	160 cm	147 cm
Heads	1000	1000	1 m${}^{3}$	84.7%	82.5%	84 cm	82 cm
Office	6000	4000	7.5 m${}^{3}$	86.8%	86.8%	188 cm	192 cm
Pumpkin	4000	2000	5 m${}^{3}$	83.4%	84.8%	220 cm	214 cm
RedKitchen	7000	5000	18 m${}^{3}$	88.7%	85.9%	188 cm	195 cm
Stairs	2000	1000	7.5 m${}^{3}$	86.2%	86.0%	189 cm	177 cm

**Table 2 sensors-22-06971-t002:** Characteristics of *12 Scenes* dataset.

Scene		Frames		Volume	Ratio of Valid Depth		Mean of Valid Depth	
		Train	Test		Train	Test	Train	Test
Apt1	Kitchen	744	357	33 m${}^{3}$	89.7%	90.4%	94 cm	97 cm
	Living	1035	493	30 m${}^{3}$	94.3%	88.2%	140 cm	137 cm
Apt2	Bed	894	244	14 m${}^{3}$	94.7%	94.5%	118 cm	117 cm
	Kitchen	782	230	21 m${}^{3}$	92.9%	92.5%	108 cm	104 cm
	Living	731	359	42 m${}^{3}$	91.2%	90.5%	138 cm	133 cm
	Luke	1370	624	53 m${}^{3}$	92.6%	91.9%	119 cm	123 cm
Office1	Gates362	3540	386	29 m${}^{3}$	90.8%	91.0%	130 cm	123 cm
	Gates381	2950	1053	44 m${}^{3}$	91.7%	91.4%	116 cm	113 cm
	Lounge	933	327	38 m${}^{3}$	94.8%	94.6%	152 cm	155 cm
	Manolis	1623	807	50 m${}^{3}$	89.7%	90.0%	116 cm	120 cm
Office2	5a	1001	497	38 m${}^{3}$	94.0%	94.1%	141 cm	137 cm
	5b	1441	415	79 m${}^{3}$	95.4%	95.2%	143 cm	155 cm

**Table 3 sensors-22-06971-t003:** Median translation error (cm) and rotation error (°) for various APR methods on *7 Scenes*. Bold numbers indicate the lowest error value.

Scene	PoseNet-17 [12]	Li’s [13]	MapNet [14]	Tian’s [15]	Ours
Chess	13 cm, 4.48°	28 cm, 7.05°	8 cm, 3.25°	9 cm, 4.39°	**7.9 cm, 2.67°**
Fire	27 cm, 11.3°	43 cm, 12.52°	27 cm, 11.69°	25 cm, 10.79°	**24.8 cm, 9.53°**
Heads	17 cm, 13.0°	25 cm, 12.72°	18 cm, 13.25°	**14 cm**, 12.56°	15.8 cm, **12.28°**
Office	19 cm, 5.55°	30 cm, 8.92°	17 cm, 5.15°	17 cm, 6.46°	**16.4 cm, 4.73°**
Pumpkin	26 cm, 4.75°	36 cm, 7.53°	22 cm, 4.02°	19 cm, 5.91°	**18.7 cm, 3.94°**
RedKitchen	23 cm, 5.35°	45 cm, 9.80°	23 cm, 4.93°	**21 cm**, 6.71°	22.2 cm, **4.47°**
Stairs	35 cm, 12.4°	42 cm, 13.06°	30 cm, 12.08°	**26 cm**, 11.51°	**26.0 cm, 9.92°**
Average	22.9 cm, 8.12°	35 cm, 10.22°	20.7 cm, 7.77°	**18.7 cm**, 8.33°	18.8 cm, **6.79°**

**Table 4 sensors-22-06971-t004:** Median translation error (m) and rotation error (°) for various APR methods on *Oxford RobotCar*. Bold numbers indicate the lowest error value.

Route	PoseNet-15 [10]	Tian’s [15]	Ours (Color Stream Only)
Loop	25.59 m, 15.96°	16.28 m, 7.17°	**10.49 m, 3.23°**

**Table 5 sensors-22-06971-t005:** Median translation error (cm) and rotation error (°) for four single depth streams with raw/full depth images as input and learning-based/handcrafted base poses on *7 Scenes* dataset. Bold numbers indicate the lowest error value.

Network	Depth Stream (Raw)		Depth Stream (Full)	
Base Poses	FC	Min	FC	Min
Chess	13.6 cm, 5.12°	11.4 cm, 4.35°	13.1 cm, 4.72°	**9.3 cm, 3.38°**
Fire	31.8 cm, 13.20°	25.0 cm, 11.75°	30.5 cm, 12.22°	**23.3 cm, 10.13°**
Heads	15.1 cm, 12.01°	14.9 cm, 12.48°	15.0 cm, 11.93°	**13.7 cm, 11.60°**
Office	23.7 cm, 6.79°	22.4 cm, 5.96°	23.4 cm, 6.74°	**16.1 cm, 4.71°**
Pumpkin	29.7 cm, 6.94°	27.5 cm, 5.69°	25.9 cm, 5.44°	**22.5 cm, 4.84°**
RedKitchen	36.3 cm, 7.72°	29.4 cm, 6.31°	33.7 cm, 7.27°	**20.0 cm, 5.41°**
Stairs	39.1 cm, 12.42°	35.9 cm, 12.79°	39.0 cm, **9.65°**	**31.9 cm**, 11.42°
Average	27.0 cm, 9.17°	23.8 cm, 8.48°	25.8 cm, 8.28°	**19.5 cm, 7.35°**

**Table 6 sensors-22-06971-t006:** Median translation error (cm) and rotation error (°) for single streams with different base poses on *7 Scenes*. (a) Color stream; (b) depth stream. Bold numbers indicate the lowest error value.

(a)				
Base Poses	Min	Cube	Mixed	FC
Chess	**7.9 cm, 2.78°**	8.7 cm, 3.19°	8.0 cm, 3.37°	8.9 cm, 3.39°
Fire	**24.2 cm, 9.22°**	24.7 cm, 10.46°	26.1 cm, 9.90°	27.5 cm, 9.75°
Heads	15.8 cm, 12.00°	**15.7 cm**, 12.70°	15.9 cm, 11.90°	16.6 cm, **11.83°**
Office	16.6 cm, **4.93°**	**16.2 cm**, 5.07°	17.3 cm, 5.04°	19.3 cm, 5.35°
Pumpkin	**19.4 cm, 3.87°**	21.2 cm, 4.32°	19.6 cm, 4.18°	22.5 cm, 4.48°
RedKitchen	22.0 cm, 4.93°	**21.9 cm**, 4.86°	21.9 cm, 5.59°	22.8 cm, **4.63°**
Stairs	28.7 cm, 10.76°	**25.5 cm**, 9.76°	29.7 cm, 10.73°	26.4 cm, **9.51°**
Average	19.2 cm, **6.93°**	**19.1 cm**, 7.19°	19.8 cm, 7.24°	20.6 cm, 6.99°
**(b)**				
**Base poses**	**Min**	**Cube**	**Mixed**	**FC**
Chess	**9.3 cm, 3.38°**	10.2 cm, 4.38°	10.0 cm, 4.37°	13.1 cm, 4.72°
Fire	**23.3 cm**, 10.13°	24.7 cm, **9.47°**	27.6 cm, 10.73°	30.5 cm, 12.22°
Heads	13.7 cm, 11.60°	**13.0 cm**, 12.30°	13.5 cm, **10.85°**	15.0 cm, 11.93°
Office	**16.1 cm, 4.71°**	16.6 cm, 5.35°	16.8 cm, 6.02°	23.4 cm, 6.74°
Pumpkin	22.5 cm, 4.84°	**21.9 cm, 4.36°**	22.1 cm, 5.19°	25.9 cm, 5.44°
RedKitchen	20.0 cm, 5.41°	19.8 cm, 5.26°	20.7 cm, 5.45°	33.7 cm, 7.27°
Stairs	**31.9 cm**, 11.42°	34.2 cm, 10.91°	34.1 cm, 10.09°	39.0 cm, **9.65°**
Average	**19.5 cm, 7.35°**	20.1 cm, 7.43°	20.7 cm, 7.53°	25.8 cm, 8.28°

**Table 7 sensors-22-06971-t007:** Median translation error (cm) and rotation error (°) for single streams with different base poses on *12 Scenes*. Bold numbers indicate the lowest error value.

Network		Color Stream		Depth Stream	
Base Poses		Min	FC	Min	FC
Apt1	Kitchen	7.7 cm, 4.39°	9.1 cm, 7.51°	**6.8 cm, 4.24°**	**6.8 cm**, 4.99°
	Living	9.2 cm, 3.53°	**8.8 cm**, 4.26°	9.0 cm, **3.48°**	9.2 cm, 3.58°
Apt2	Bed	**8.7 cm, 4.75°**	9.6 cm, 6.88°	11.0 cm, 4.87°	10.8 cm, 6.04°
	Kitchen	7.0 cm, **3.77°**	10.5 cm, 7.70°	6.8 cm, 4.23°	**6.6 cm**, 6.98°
	Living	9.5 cm, **3.99°**	10.9 cm, 5.37°	**9.0 cm**, 4.23°	8.9 cm, 6.76°
	Luke	12.5 cm, 4.38°	11.9 cm, 4.01°	9.8 cm, **3.57°**	**8.6 cm**, 5.79°
Office1	Gates362	6.4 cm, **2.32°**	6.7 cm, 2.71°	5.6 cm, 3.05°	**5.3 cm**, 2.94°
	Gates381	12.6 cm, **5.19°**	12.6 cm, 7.18°	**10.3 cm**, 5.61°	10.7 cm, 5.94°
	Lounge	8.3 cm, **3.42°**	10.7 cm, 6.51°	**6.8 cm**, 4.12°	8.3 cm, 4.98°
	Manolis	8.5 cm, 3.97°	10.3 cm, 5.79°	**8.3 cm, 3.60°**	9.5 cm, 4.79°
Office2	5a	**8.9 cm**, 4.43°	12.8 cm, 5.39°	10.7 cm, **4.31°**	11.3 cm, 6.24°
	5b	10.2 cm, 3.09°	10.4 cm, **2.96°**	**7.7 cm**, 3.01°	9.5 cm, 3.80°
Average		9.1 cm, **3.94°**	10.4 cm, 5.52°	**8.5 cm**, 4.03°	8.8cm, 5.24°

**Table 8 sensors-22-06971-t008:** Median translation error (cm) and rotation error (°) for different network architectures on *7 Scenes*. Bold numbers indicate the lowest error value.

Network	Color Stream	Depth Stream	Dual Stream
Base Poses	Min	Min	Min
Chess	**7.9 cm**, 2.78°	9.3 cm, 3.38°	**7.9 cm, 2.67°**
Fire	24.2 cm, **9.22°**	**23.3 cm**, 10.13°	24.8 cm, 9.53°
Heads	15.8 cm, 12.00°	**13.7 cm, 11.60°**	15.8 cm, 12.28°
Office	16.6 cm, 4.93°	**16.1 cm, 4.71°**	16.4 cm, 4.73°
Pumpkin	19.4 cm, **3.87°**	22.5 cm, 4.84°	**18.7 cm**, 3.94°
RedKitchen	22.0 cm, 4.93°	**20.0 cm**, 5.41°	22.2 cm, **4.47°**
Stairs	28.7 cm, 10.76°	31.9 cm, 11.42°	**26.0 cm, 9.92°**
Average	19.2 cm, 6.93°	19.5 cm, 7.35°	**18.8 cm, 6.79°**

**Table 9 sensors-22-06971-t009:** Median translation error (cm) and rotation error (°) for different network architectures on *12 Scenes*. Bold numbers indicate the lowest error value.

Network		Color Stream	Depth Stream	Dual Stream
Base Poses		Min	Min	Min
Apt1	Kitchen	7.7 cm, 4.39°	**6.8 cm**, 4.24°	7.0 cm, **4.17°**
	Living	9.2 cm, 3.53°	9.0 cm, **3.48°**	**8.3 cm**, 3.74°
Apt2	Bed	8.7 cm, **4.75°**	11.0 cm, 4.87°	**7.6 cm**, 4.92°
	Kitchen	7.0 cm, 3.77°	**6.8 cm**, 4.23°	7.1 cm, **3.12°**
	Living	9.5 cm, **3.99°**	**9.0 cm**, 4.23°	**9.0 cm**, 4.16°
	Luke	12.5 cm, 4.38°	**9.8 cm, 3.57°**	11.2 cm, 4.42°
Office1	Gates362	6.4 cm, **2.32°**	**5.6 cm**, 3.05°	6.3 cm, 2.41°
	Gates381	12.6 cm, **5.19°**	**10.3 cm**, 5.61°	11.6 cm, 5.83°
	Lounge	8.3 cm, **3.42°**	**6.8 cm**, 4.12°	8.9 cm, 3.69°
	Manolis	8.5 cm, 3.97°	**8.3 cm**, 3.60°	8.8 cm, **3.57°**
Office2	5a	**8.9 cm**, 4.43°	10.7 cm, 4.31°	10.6 cm, **3.29°**
	5b	10.2 cm, 3.09°	**7.7 cm**, 3.01°	10.6 cm, **2.79°**
Average		9.1 cm, 3.94°	**8.5 cm**, 4.03°	8.9 cm, **3.84°**

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
