# Peer review of "Absolute Camera Pose Regression Using an RGB-D Dual-Stream Network and Handcrafted Base Poses"

_sensors, 2022, doi:10.3390/s22186971_

Round 1
Reviewer 1 Report
A modified version of absolute pose regression (APR) for camera localization has been proposed. My main concern is about the handcrafted base-poses incorporated in APR, which is the main contribution of the paper. Please see the attached document for detailed comments.

Author Response
Point 1: My main concern is about the handcrafted base poses incorporated in APR, which is the main
contribution of the paper. The statement on lines 21-216 indicates that base poses are no more
handcrafted. Please give more detail about how you determine/learn hand-crafted base poses
and use them.
Response 1: The selection of the handcrafted base poses is introduced in section 3.6.2 “Selection of Handcrafted Base Poses”. The way how to use the hancreated base pose can be found in the section 3.3 “Network Architecture” and section 3.5 “Training Mechanism”.
Point 2: Line 173: What do you mean by the distribution of a camera?
Response 2: We mean the ground truth camera trajectories used for training. We have modified the line 173 to “APR is a data-driven camera localization approach by which the distribution of the ground truth camera trajectories used for training can limit base poses learned from the training data.”
Point 3: Lines 177-180: It is a big claim. It must be validated.
Response 3: The statement in line 177-180, “However, since the training trajectory does not necessarily include diverse directions, redundant base poses may not cover the space outside the training trajectory. In other words, the APR model that adopts base poses based on learning has no generalizability.”
According to the quantitative results shown in table 3 and 4, we can say that our model’s generaliability is better than the previous methods because our model’s localization error is smaller than the previous methods.
Point 4: Line 191: What is the specification of these layers?
Response 4: Each of these FC layers is a single-layer with full connection.
Point 5: Line 201: What do mean by adjacent images?
Response 5: We mean the temporally adjacent images. We have modified the line 201 to “the temporally adjacent images”.
Point 6: Lines 227-228: How the hyper-parameters are fixed?
Response 6: The statement in line 227-228, “The β and Δt hyperparameters are set to 3 and 10 for all scenes, γ is set to 10 for indoor and 1 for outdoor datasets.” When we train the model, β, Δt, and γ are fixed by setting them as constant values.
Point 7: Line 246-247: This statement is not valid and contradicts the one on line 251.
Response 7: The statement in line 246-247, “This requirement can only be met by mapping the features to a linear embedding without an activation function.” The statement in line 251, “We adopt the leaky rectified linear unit (leaky ReLU) as the activation function for nonlinear embeddings to streamline network convergence.”
The statement in line 246-247 is under the premise that only three orthonormal vectors in the positive direction are used as base poses. We use six orthonormal vectors (three in positive and three in negative directions) used as the base poses.
Point 8: Figure 5a: Are these randomly selected? Why these are only positive contrary to the statement on Line 250.
Response 8: These six orthonormal vectors are not randomly selected. They are (1, 0, 0), (0, 1, 0), (0, 0, 1), (-1, 0, 0), (0, -1, 0), and (0, 0, -1). According to the values of on the axes, they are not all positive. Three of them are positive, and the other ones are negative. The origin is on the center of the graph.
Point 9: Figure 5b, c: These are redundant base-poses? Why this number is selected?.
Response 9: Yes, the base poses in Figure5b,c are redundant base poses. The reason that the number 2,048 is selected is that the dimension of the feature verctor after ResNet-18 is 2,048. We follow the Tian’s method (3D scene geometry-aware constraint for camera localization with deep learning) to choose the dimension of the feature vector after ResNet-18.
Point 10: Table 3: Please give the detail on whether you used the data of a specific scene only to train the model for that scene.
Response 10: We do, and the reason for doing it is that the proposed method is an APR method. APR method is one kind of the single-shot camera localization methods. Single-shot camera localization estimates the camera pose from a single image related to a previously visited scene. It has two phases. In phase 1, a model (e.g., 3D point cloud, neural network) is first built based on a set of images. In phase 2, the camera pose related to the model is estimated based on a single image.
Point 11: The size of training data looks small. How did you tackle the overfitting problem?
Response 11: We use handcrafted base poses to avoid the localization results overfitting to the training data. It is because that the scene may be very large (like RobotCar dataset) that the training trajectories cannot cover the entire scene.

Reviewer 2 Report
This paper has proposed a novel camera pose estimation method targeting the RGB-D environment.
My major concern is the proposed network, where the authors have designed their method without considering the distinct attribute of the D channel.
Thus, the proposed method is generic and can also be applied to a normal RGB environment.
Also, the overall paper writing MUST be improved significantly because I have encountered massive undefined mathematical symbols.
Author Response
Point 1: My major concern is the proposed network, where the authors have designed their method without considering the distinct attribute of the D channel.
Thus, the proposed method is generic and can also be applied to a normal RGB environment.
Also, the overall paper writing MUST be improved significantly because I have encountered massive undefined mathematical symbols.
Response 1: The reason for using the same network backbone is to let the two-stream models in our network share the same weights, which our experiments had shown better network performance. Also, we have carefully checked all the mathematical symbols in this paper, and have not found any undefined symbol. Please kindly point out the undefined mathematical symbols you found, so that we can revise the paper accordingly.

Author Response
Point 1: Indeed manuscript presented well with good technical and experimental design, but introduction section needs additional discussion regarding specific challenges that involved in APR methods’ based on base poses learned from training data alternative existing methodologies. Also actual inspiration behind choosing handcrafted feature against learned feature for base poses using in APR method.
Response 1: We have revise the introduction section for more information about why using handcrafted base poses is better than using learning-based base poses. We also adds the description of using learning-based base poses has some performance limitation.
